# Design and Implementation of Three-Channel Drainage Pipeline Ground Penetrating Radar Device

**DOI:** 10.3390/s23094525

**Published:** 2023-05-06

**Authors:** Maoxuan Xu, Feng Yang, Rui Yan

**Affiliations:** 1School of Mechanical Electronic and Information Engineering, China University of Mining and Technology (Beijing), Beijing 100083, China; yangf@cumtb.edu.cn; 2Beijing Drainage Group Co., Ltd., Beijing 100044, China; yanr@bdc.cn

**Keywords:** ground penetrating radar, drainage pipeline detection, inspection device

## Abstract

In order to solve the current problems that conventional video inspection can only detect, as an internal pipeline defect and drainage pipeline radar inspection device detects in a single direction and at radar frequency in water pipeline defect detection, a three-channel drainage pipeline ground penetrating radar (GPR) inspection device was designed and developed, the assembly and commissioning of the device prototype were completed, and an actual engineering test application was carried out. Focusing on the problem that the detection direction and depth of the single-channel detection device are limited, a three-channel drainage pipeline GPR inspection device is designed to realize the synchronous detection of the inside of the pipeline, the pipeline body, and the external environment of the pipeline, improving the detection depth and efficiency. According to the design scheme of the three-channel drainage pipeline GPR inspection device, the assembly of the device prototype was completed. The device contains three radar channels, the top of the main frequency of the antenna is 1.4 GHz, the two sides are 750 MHz, the video camera has a pixel count of 4 million, and the positioning accuracy is less than 1 mm, the waterproof grade is IP68, the detection accuracy of pipe deformation (slope) is 0.1°, the detection depth outside the pipe is 1.2 m, and the detection accuracy of corrosion thickness is 15 mm. In a practical application of the device, the Jianguomenqiao sewage pipeline in Beijing, China, was tested, resulting in the discovery of 87 defects, including 39 loose soil areas at the bottom of the pipe exterior, 40 void areas, and 8 cavities.

## 1. Introduction

The urban underground drainage network is widely distributed. The leakage and damage to an underground drainage pipeline will lead to ground collapse, and the blockage of pipeline sludge will lead to urban drainage blockage and urban waterlogging, threatening the safety of urban operations [1]. Drainage pipes in municipal underground pipelines are often laid in the lowest layers of the ground, and broken and leaky drainage pipes take away the surrounding soil, leaving voids around the pipes and causing upper roads to collapse [2]. Therefore, inspection of drainage pipe quality and voids is essential to combat urban road subsidence.

In response to the above problems, many scholars have carried out a lot of research on the detection techniques and equipment for the urban underground in recent years. closed-circuit television (CCTV) technology to detect defects on the internal surface of pipes has been used for more than 40 years [3]. It is mainly used in sewers and storm water pipes. CCTV mounted on a robot inside a crawler pipe allows the inspector to detect any structural defects by analyzing the returned video and recording the defects through a snapshot of the internal surface [4].

The main conventional in-pipe video inspection equipment currently available is the S300 pipeline image inspection robot from SROD company in China [5], which uses a steering track to traverse the pipeline for video recording, transmits data via cable, and displays images on an intelligently controlled image recording terminal. The robot is equipped with a high-definition camera on its lifting wall and is mainly used for video inspection of pipes with a diameter of 300~2000 mm, enabling the recording of pictures and location information of pipeline defects. The US RedZone company’s SOLO [6], a small autonomous traveling drain inspection robot, is equipped with one panoramic spherical lens at the front and rear, which can record 360° video in the pipe and is suitable for 200~300 mm pipe diameter waterless pipes.

CCTV requires dewatering, debris removal, and a human operator for sewer inspection, which makes this approach costly and time consuming. Moreover, CCTV is unable to detect any voids in the backfill soil and is often carried out after a structural failure or blockage report was made [7].

In recent years, many scholars have conducted extensive research on ground penetrating radar (GPR) technology, a non-destructive, high-resolution electromagnetic (EM) technique that was primarily designed to investigate the subsurface detection of buried utility infra-structure, including pipes [8]. For example, detecting leaks in drainage pipes [9], cavity defects in the shallow subsurface [10], the enclosed underground sewerage pipe spaces [11], and early cracking of cement concrete pavements [12,13]. Techniques for detecting structural deterioration of water mains can determine their current condition, including their structural health, impact on water quality, and hydraulic capacity [4,14]. GPR can potentially identify leaks in buried water pipes either by detecting underground voids created by the leaking water or by detecting anomalies in the depth of the pipe as the radar propagation velocity changes due to soil saturation with leaking water [15]. The GPR technique was also applied to determine the degree of internal leaching of hydroxides in asbestos-cement (AC) pipes [16].

Pipe penetrating radar (PPR) is the in-pipe application of ground penetrating radar (GPR). PPR is an advanced method for detecting defects in drainage pipes. In comparison to more common condition inspection methods like CCTV, PPR provides quantitative, repeatable, and actionable data to the inspector [17].

Relevant inspection equipment include: The GPR300 single-channel pipeline radar robot for drainage pipes of 300 mm diameter developed by the China University of Mining and Technology (Beijing) [18]. It has an industrial grade HD camera and a 1.5 G GPR antenna, which is used to detect corrosion, voids, and cracks at the top of pipes. The ACPS asbestos cement pipe scanner from Sewer VUE Technology Canada is equipped with a high-frequency antenna (1.6 GHz or 2.3 GHz) for wall thickness and void detection in pipes up to 250~450 mm in diameter [19]. Another of the company’s dual-channel equipped MSI multi-sensor inspection robots uses video, laser, and pipe radar technology to inspect underground pipes [17]. The radar data are collected via two independent channels and are suitable for wall thickness inspection of pipes with diameters from 500 to 1500 mm and for locating voids outside the pipe wall. However, it is too large to detect drainage pipes with a diameter of 400~600 mm, and the limited rotation angle of the two antennas does not allow it to cover more detection areas.

The existing detection methods have problems such as independent operation and low detection efficiency, which make it difficult to meet the needs of the detection of drainage pipeline defect engineering. Therefore, it is very important to increase the research on the equipment and technology related to drainage pipe defect detection equipment and technology to improve the depth and efficiency of drainage pipe defect detection.

Based on the research idea of combining GPR with pipeline robots, an inspection device was developed to enable pipeline robots carrying GPR to achieve simultaneous detection of internal, main body, and external environmental defects of the pipeline.

## 2. Three-Channel Drainage Pipe GPR Inspection Device Design

### 2.1. Structural Design of Three-Channel Drainage Pipeline GPR Inspection Device

Conventional drainage pipe inspection equipment only has video sensors, which can only be used for internal pipe defect detection; there are video sensors combined with single-channel ground-penetrating radar detection equipment, which has limited coverage and depth of detection. In order to solve this problem, the research and development of the three-channel drainage pipeline ground-penetrating radar device was carried out, and three channels were used for operation in three different directions of the drainage pipeline to realize the synchronous detection of the interior of the drainage pipeline, the pipeline body, and the external environment of the pipeline.

The three-channel drainage pipeline GPR inspection device includes a waterproof pipeline robot device, a GPR detection device, an embedded acquisition and control system, an embedded interface circuit, and a host computer, as shown in Figure 1.

The waterproof pipeline robotics unit runs inside the pipe to detect defects in the pipe. It is mainly composed of a robot body, a robot arm, and a robot walking system. The robot arm adopts an adaptive design to meet the detection needs of complex conditions such as deformation in the pipeline.

Drainage pipes in China are currently dominated by concrete pipes. As the pipes get older, microorganisms in the sewage inside the pipes produce hydrogen, which causes corrosion of the concrete pipes. As corrosion is usually most severe at the top of the pipeline, GPR technology is used to detect the depth of corrosion and determine defects in the pipeline. The depth and accuracy of detection depends on the frequency of the GPR antenna, as shown in Table 1.

The three-channel GPR inspection device is composed of three channels, the angle between the three channels is 120°, and each channel is composed of a video camera system and an antenna system. The tube top radar antenna system is the first channel, the GPR antenna main frequency is 1.4 GHz for corrosion depth and wall thickness inspection of the pipe tops, the left tube bottom radar antenna system is the second channel, the right tube bottom radar antenna system is the third channel, the main frequency of the left tube and right tube antenna is 750 MHz for the detection of external defects on the bottom of the tube, as shown in Figure 2.

### 2.2. Three-Channel Drainage Pipeline GPR Hardware Design

The three-channel drainage pipeline GPR control system is composed of a GPR detection device and an embedded acquisition control system. It is composed of a radar acquisition control module, a gyroscope module, three radar transmitters, three radar receivers, three arc-shaped transceiver antennas, and a power supply module, as shown in Figure 3.

The three-channel drainage pipeline GPR acquisition and control system adopts the embedded design of ethernet technology. The system is based on the client/server (C/S) network architecture, and the network transmission protocol is the UDP protocol. The lower computer, the server, uses the Arm Cortex-M7 chip as the core to build an embedded acquisition and transmission platform. The server uses the RT Thread embedded real-time operating system. The application program is based on the embedded lightweight TCP/IP protocol (LwIP) protocol stack RAM interface development. The controller uses the data transceiver to collect and convert radar transmitted wave signals to realize the transmission of radar data and control commands.

The radar acquisition control module is composed of a main control microcontroller unit (MCU) chip, an analog-to-digital (A/D) conversion circuit, a digital-to-analog (D/A) conversion circuit, and a step delay circuit.

The radar acquisition control module is composed of a radar transmitter and a radar receiver. The radar antenna converts electromagnetic wave signals into electrical signals. The radar transmitter is used to generate electromagnetic wave pulses with extremely short rise times, and consists of a regulated power supply, a multi-stage avalanche circuit, a trigger pulse forming circuit, and a high-voltage module. The radar receiver is used to receive electromagnetic wave signals, and is composed of a trigger pulse forming circuit, a low-frequency avalanche circuit, a double pulse generator, a high-frequency sampling head, a high-frequency amplifier, an integrator, and a feedback circuit.

The gyroscope module adopts the MPU9250 gyroscope, which is a six-axis integrated motion processing component, which can avoid the inter-axis difference when combining the gyroscope and the accelerator and saves packaging space. The radar antenna detects the angle by using the gyroscope module to obtain the attitude information. The MPU9250 angular velocity full grid sensing range is ±250, ±500, ±1000, and ±2000°/s, which can accurately track fast and slow motions. The accelerometer sensing range is controllable, the sensing range is ±2 g, ±4 g, ±8 g, and ±16 g.

The photoelectric conversion module is designed based on a full gigabit fiber optic transceiver, with two 10/100/1000 M adaptive network ports and two 1.25 G uplink optical ports. All ports adopt non-blocking and full line-speed data packet forwarding. The network port supports automatic MDI/MDIX identification and flipping, has adaptive full-duplex/half-duplex function, supports 2048-byte frames, and supports a maximum of 10 Kbyte ultra-long data packets. The network transmission distance is extended from the limit distance of 100 m of copper wire to 120 km. The optical port single-mode wavelength is 1310/1550 nm, and the DFB wavelength is 1260~1610 nm.

The power module adopts the URA_YMD-15WR3 module and the URB_YMD-15WR3 module. The URA_YMD-15WR3 module can convert the external power supply 48 V to +12 V and −12 V. The URB_YMD-15WR3 module can convert the external power supply 48 V to +5 V.

The power supply module supplies power to the radar acquisition control module, and the main control MCU chip of the radar acquisition control module transmits the radar signal to the host computer through the photoelectric conversion module. The trigger signal of the radar transmitter and the trigger signal of the radar receiver are generated by a DA conversion circuit, and the A/D conversion circuit collects the radar signal received by the radar receiver. The gyroscope module collects the detection angle of the radar antenna and transmits it to the host computer through the radar acquisition control module. The host computer combines the data to calibrate the radar detection angle to improve the accuracy of the detection results.

### 2.3. Overall Design of GPR Data Acquisition for Three-Channel Drainage Pipeline

The overall process of GPR data acquisition for three-channel drainage pipes is shown in Figure 4. The upper computer serves as the sending end to send commands to the lower computer at the receiving end; after the lower computer receiving end obtains the command information of the upper computer, it needs to analyze it. There are two types of modes: the single-channel sampling mode and the continuous multichannel sampling mode designed in this area, and the single-channel buffer or multichannel sampling circular queue buffer is selected. According to the single sampling and continuous sampling commands, the sampling process is executed, and finally, the collected channel number information is sent to the host computer to complete the data sending process.

## 3. Realization of Three-Channel Drainage Pipeline GPR Inspection Device

### 3.1. Prototype Assembly of Three-Channel Drainage Pipeline GPR Inspection Device

According to the design drawing of the three-channel drainage pipeline GPR inspection device (as shown in Figure 2), the assembly of the three-channel drainage pipeline detection device is completed, and the model machine photo is shown in Figure 5.

The three sides of the main body of the three-channel drainage pipeline GPR inspection device are connected to the receiving and transmitting antenna box, the plastic upper cover of the antenna box is fixed on the metal plate at its bottom with screws, the radiation surface and absorbing material of the GPR transmitting and receiving antenna are contained in the plastic upper cover, the metal plate at the bottom of the antenna box and the corresponding position of the main body are installed with a limited base, the metal rod is used to form the rocker to connect the two parts, there are two plastic directional wheels on both sides of the metal plate at the bottom of each antenna box, and a three-side directional wheel forms the fulcrum and attaches to the inner wall of the pipe to make the main body move in the direction of the pipe. A U-shaped metal handle is installed on one side of the main body, and steel wire rope is installed on the camera side, which is fixed to the main body with screws for traction and handling.

The support wheels of the curved antennas on the left and right sides of the three-channel drainage pipe GPR inspection device are McNamee wheels that can rotate omnidirectionally, allowing the robot to spiral forward in the pipe and thus measure the GPR signal in the circumferential direction. The top curved antenna is fitted with a guide wheel structure in the forward direction. When the robot rotates itself during the inspection process, the internal inertial navigation sensor acquires the angle of rotation and the master control chip controls the reverse rotation of the guide wheel, allowing the PPR to correct the deflection, thus ensuring the accuracy of the inspection angle.

The main body of the three-channel drainage pipeline GPR inspection device includes the acquisition control circuit, camera, radar receiver, radar transmitter, etc.

### 3.2. Technical Specifications Test of Three-Channel Drainage Pipeline GPR Inspection Device

#### 3.2.1. Positioning Accuracy

In order to achieve high-precision mileage positioning of the pipeline robot, the 1000BZ-05CG21 pulse encoder is used to sample its mileage information. The circumference of the ranging wheel is 330 mm, the number of pulses is 1000, and the maximum mileage positioning accuracy is 0.33 mm. As the robot travels through the tube, the distance measuring wheel and encoder, which are fixed to the line carriage, are rotated by a transmission cable to obtain mileage information.

#### 3.2.2. Sensitivity Test of the Gyroscope Linear Acceleration

In order to realize the high-precision spatial positioning and attitude confirmation of the pipeline robot, the MPU9250 electronic gyroscope is used to sample its linear acceleration, with a three-axis angular velocity sensor of ±2000°/s and a three-axis accelerator of ±16 g. The sensitivity of the linear acceleration of the gyroscope is 0.06°/s/g.

#### 3.2.3. Video Camera Pixel Test

The video camera uses a SONY VRS-MH8100 high-definition camera, 1/3 CMOS, 10.5 times optical zoom, the effective pixel is 4.08 million, and the camera resolution of this model is 1080p full HD.

#### 3.2.4. Slope Test Index

The slope is obtained from the pipeline model, and then the slope estimation parameters at different locations are obtained. The slope of the pipeline model is −3.25°, the actual test slope is between −3.13° and −3.35°, and the slope accuracy is 0.1°.

#### 3.2.5. Radar Detection Index Test

The number of channels of the test device is three. The detection depth test is carried out in a known pipeline, and the 400–600 mm pipeline robot detection image is obtained. The depth of the cavity detection is more than 1.2 m, as shown in Figure 6. It indicates that the detection depth can reach 1.2 m.

This three-channel drainage pipeline GPR device is dedicated for use in pipe defect detection. To illustrate the performance of this system, the comparison of the key parameters between this system and the other GPR systems (ACPS), which can be used to detect defects in 400~600 mm drainage pipes, is shows in Table 2.

## 4. Three-Channel Drainage Pipeline GPR Detection Technology Application

Based on the developed three-channel drainage pipeline GPR inspection device, an experimental application of urban drainage pipeline defect detection project was carried out. The following is the test case of Jianguomenqiao sewage pipeline in Beijing, China.

### 4.1. Detection Purpose and Detection Location

The purpose of this inspection is to inspect the structural and functional conditions of the drainage pipeline, including the soil anomalies, cracks, bulges, corrosion, and other defects around the pipeline, as well as the pipeline joints, pipeline deflection, and other conditions. The inspect standards are based on Beijing Drainage Group’s corporate standards: “Drainage Canal Structure Rating Standards” (JS001-GW05-2012), “Drainage Canal Function Rating Standards” (JS002-GW05-2012), and Urban Engineering Geophysical Detection Standards (CJJ/T 7-2017).

The test area is located at Jianguomen Bridge, Beijing, China. The observation line is located almost in the northerly direction. There are eight pipe sections on the observation line with a total length of 334.3 m, numbered from north to south. The pipelines are burial depth 3 m made of concrete pipes with a diameter of 400~600 mm, as shown in Figure 7.

### 4.2. Data Acquisition Process

The three-channel drainage pipeline radar detection device is used for pipeline defect detection. The number of antenna channels is three, the antenna frequency at the top is 1.4 GHz, and the antenna frequency at both sides is 750 MHz. One CCTV industrial camera is used.

The 1.4 GHz radar antenna is used to detect the corrosion depth and wall thickness at the top of the pipeline with a sampling time window of 10 ns. A 750 MHz radar antenna is used to detect voids on the outside of the pipeline with a sampling time window of 30 ns. The instrument is triggered by a high-precision rangefinder wheel in the line trolley with a trace interval of 0.5 mm.

The acquisition process of the three-channel drainage pipeline detection device is shown in Figure 8. The data acquisition includes the environmental and geological background investigation of the construction site, instrument installation and commissioning, survey line layout, parameter setting of the radar detection system for the three-channel drainage pipeline, and field parameter commissioning. It is worth noting that the antenna should be as close to the inner wall of the pipeline as possible during the detection.

### 4.3. Drainage Pipeline Defect Detection Results

A total of 87 defects were found during the drainage pipeline defect detection, including 39 loose soil areas at the bottom of the pipeline, 40 void areas, and 8 cavities.

#### 4.3.1. Void Defect Detection

A void refers to the gap between the outside of the pipe wall and the structural plane. The void area is often small vertically and develops horizontally along the structural plane. The void is often filled with air or rich water.

The typical void defect GPR image is shown in Figure 9. The void interface has strong reflection, which is distributed in an approximate horizontal band with multiple reflection signals. The detected void is located on the left side of the survey line with the pipe section number of 25,822~28,384 and starting and ending mileages of 9.9~11.8 m.

#### 4.3.2. Loose Defect Detection

Looseness refers to a loose soil layer around the pipeline with insufficient bearing capacity. The loose defect is mainly caused by the substandard compactness of the backfill soil when the excavation part is backfilled during the pipeline placement.

A typical loose defect GPR image is shown in Figure 10. The loose interface is a strong reflection, and the isophase axis is discontinuous, staggered, and disordered. The detected looseness is located on the left side of the survey line with the pipe section number of 25,822~28,384 and starting and ending mileages of 17.5~19.5 m.

#### 4.3.3. Cavity Defect Detection

A cavity refers to the separation between the structural layer and the soil foundation. The scale of the cavity is large, and the soil mass in the internal area of the structural layer is missing.

A typical cavity defect GPR image is shown in Figure 11. The boundary surface of the cavity has a strong reflection, which is a regular or irregular hyperbolic waveform and is characterized by a solitary phase. The number of the detected pipe section is 25,822~28,384, and the starting and ending mileages are 52.7~53.3 m, as located on the right side of the survey line.

## 5. Conclusions

In this paper, a three-channel drainage pipe radar (GPR) unit for drainage pipe defect detection is designed and tested, incorporating multi-sensor technologies such as closed-circuit television (CCTV) and inertial navigation units (IMU). Compared to other pipeline inspection systems, this system is specifically adapted for pipeline defect detection and has the following design features: three-channel operation effectively improves the survey efficiency, achieving reflection radar data with different frequencies from three angles in one single pass; it can realize the synchronous detection of the interior, body, and external environments of the pipeline.

The experimental application results verify that the design is a relatively fast and effective non-destructive testing method that does not interrupt the operation of the drainage pipe network. The corrosion detection accuracy of the top of the pipe is 15 mm and the detection range of the cavity outside the pipe is 1.2 m, solving the problem of limited detection angle and depth.

For the next step of development, we will investigate situations in which the device may slip or rotate during the inspection process inside the pipe, and improve the control algorithm to correct its own rotation to improve the accuracy of the detection angle of this device.

## Figures and Tables

**Figure 1 sensors-23-04525-f001:**
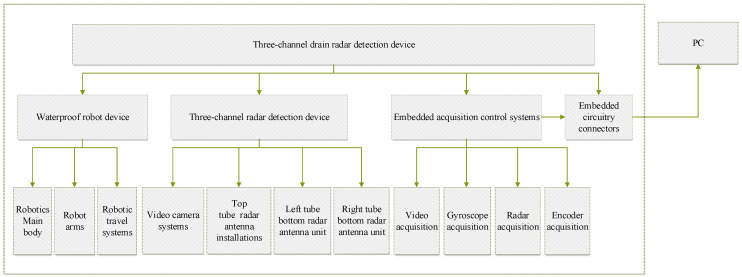
The composition of the three-channel drainage pipe GPR inspection device.

**Figure 2 sensors-23-04525-f002:**
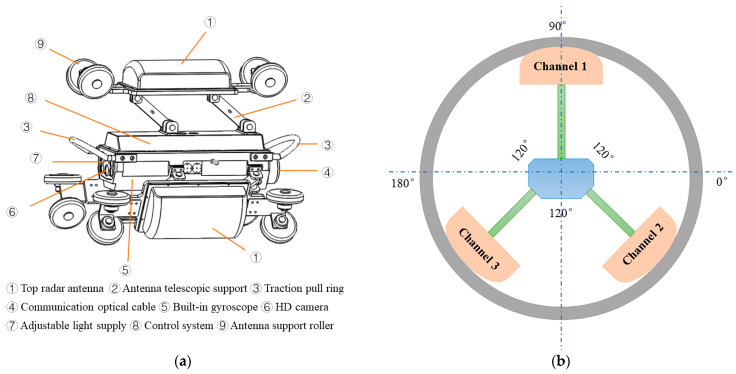
Schematic diagram of three-channel drainage pipe GPR inspection device. (**a**) Device structure diagram; (**b**) three channel directions.

**Figure 3 sensors-23-04525-f003:**
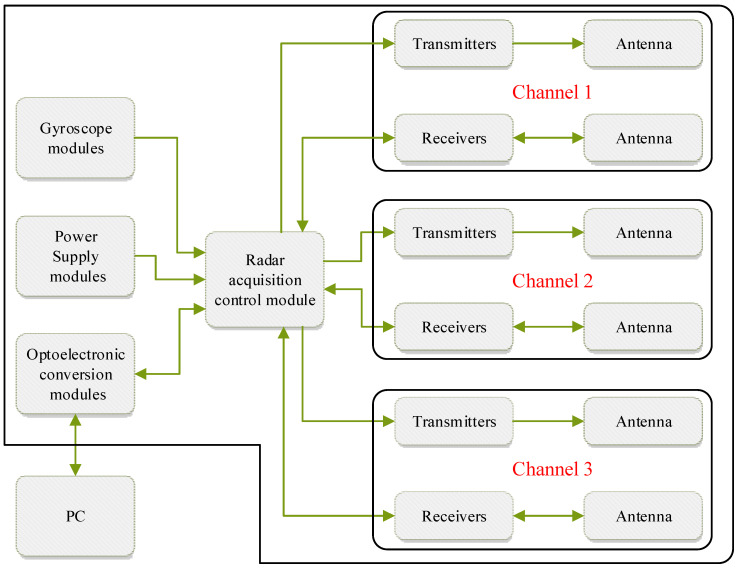
Drainage pipe GPR acquisition control system hardware framework diagram.

**Figure 4 sensors-23-04525-f004:**
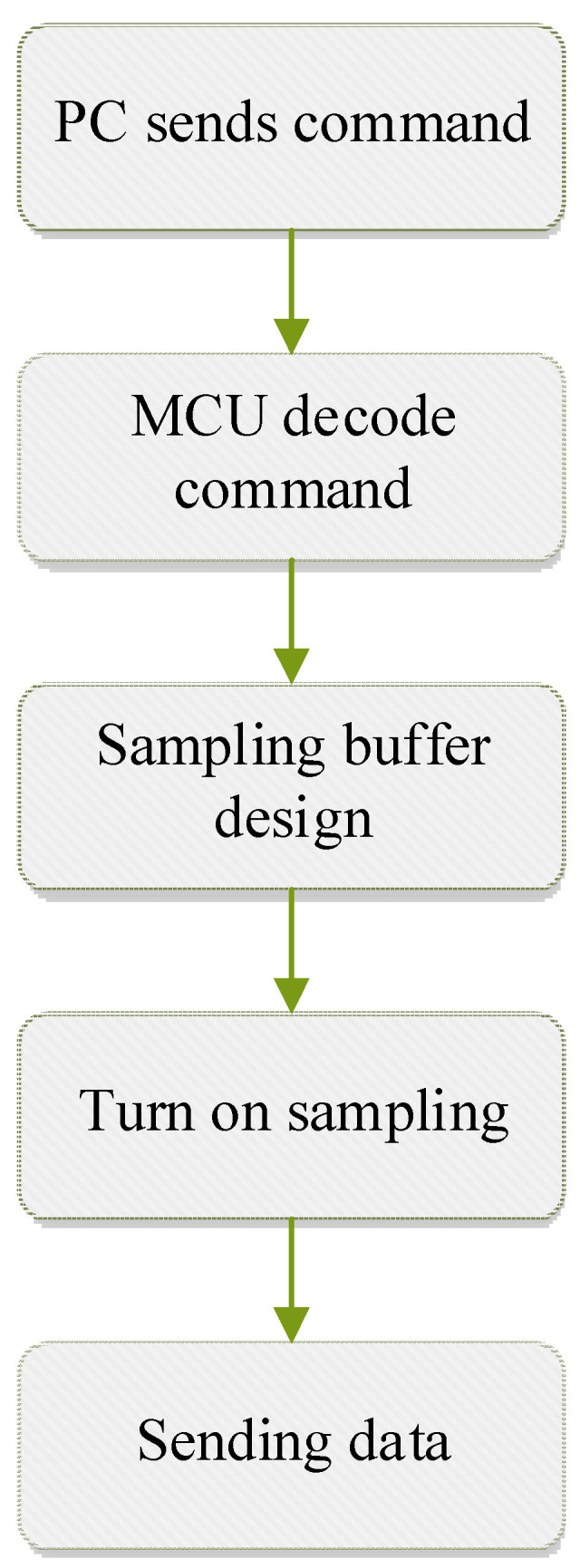
General process flow chart of GPR data acquisition for drainage pipeline.

**Figure 5 sensors-23-04525-f005:**
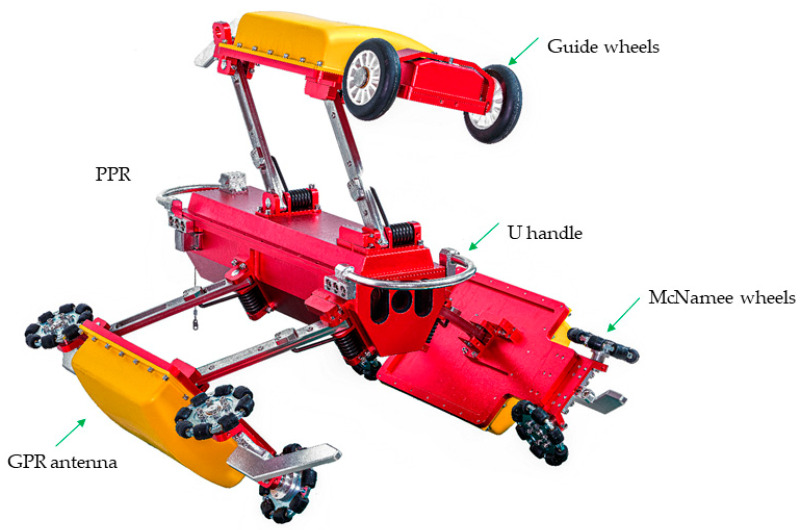
Three-channel drainage pipeline GPR inspection device model machine.

**Figure 6 sensors-23-04525-f006:**
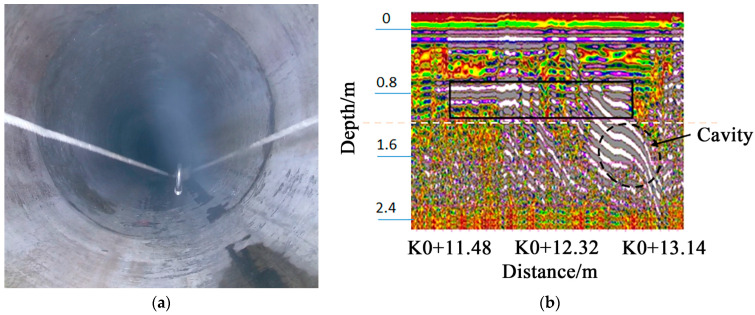
Cavity detection depth test. (**a**) The tested pipes; (**b**) GPR detection images.

**Figure 7 sensors-23-04525-f007:**
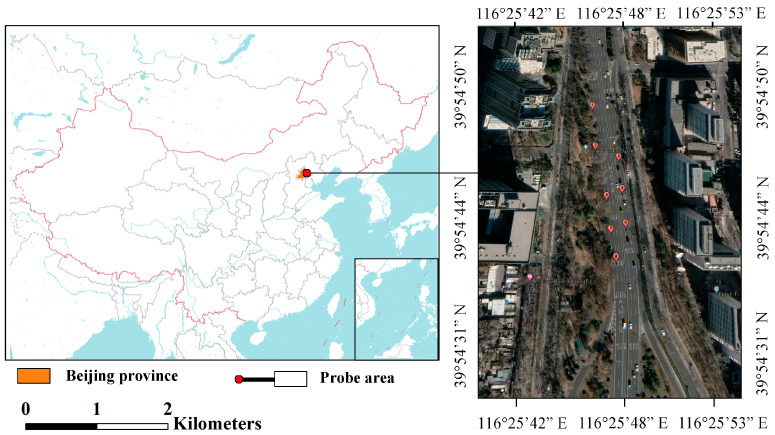
Line layout of the pipeline detection area.

**Figure 8 sensors-23-04525-f008:**
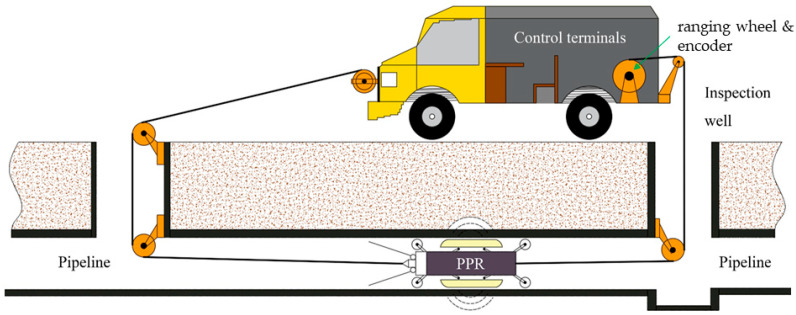
The acquisition process of the three-channel drainage pipe inspection.

**Figure 9 sensors-23-04525-f009:**
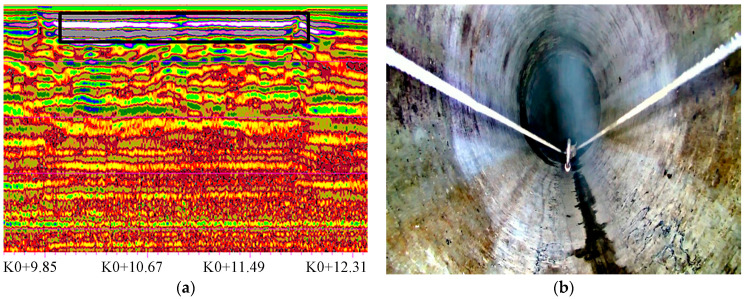
Void defect GPR image. (**a**) GPR profile; (**b**) video image.

**Figure 10 sensors-23-04525-f010:**
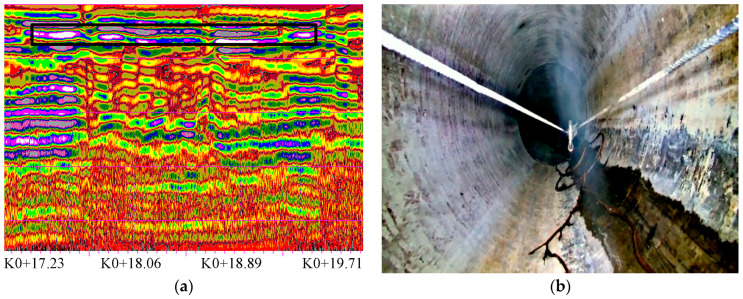
Loose defect detection. (**a**) GPR profile; (**b**) video image.

**Figure 11 sensors-23-04525-f011:**
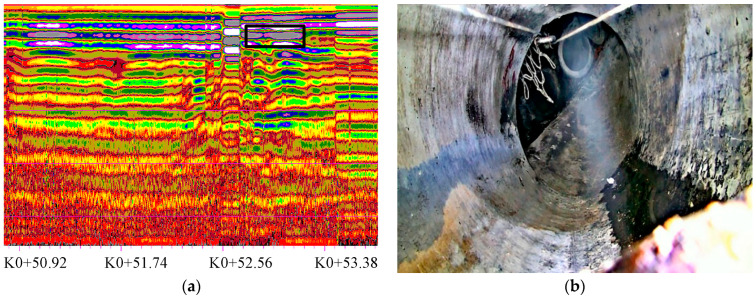
Cavity defect GPR image. (**a**) GPR profile; (**b**) video image.

**Table 1 sensors-23-04525-t001:** Horizontal resolution of high-frequency GPR antennas in concrete.

Depth (cm)	Horizontal Resolution (cm)		
	500 Mhz	1000 MHz	1500 MHz
5	7.1	5.0	4.1
10	10.0	7.0	5.8
15	12.2	8.6	7.1
20	14.1	10.0	8.2
25	15.8	11.2	9.1
30	17.3	12.2	10.0
35	18.7	13.3	10.8
40	20.0	14.3	11.5
45	21.2	15.1	12.2

**Table 2 sensors-23-04525-t002:** Key parameter comparison of the pipeline defect detection GPR.

Type	Three-Channel Drainage Pipeline GPR Device	ACPS
Number of radar channels	3	1
Antenna frequency	Top: 1.4 GHz; Left and right: 750 MHz	1.6 GHz/2.3 GHz
Camera pixels	4 megapixels	2 megapixels
Location accuracy	<1 mm	Unspecified (±5%)
Waterproof	IP68	Yes
Accuracy of pipeline deformation measurement	0.1°	No
Pipeline external detection depth	1.2 m	0.92 m
Bottom of pipe detection capability	Yes	No

## Data Availability

Not applicable.

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
