# Peer review of "Design and Implementation of Three-Channel Drainage Pipeline Ground Penetrating Radar Device"

_sensors, 2023, doi:10.3390/s23094525_

Round 1

Reviewer 1 Report

In this manuscript, a three-channel pipe penetrating radar (PPR) is designed for inspection of the structural and functional conditions of underground drainage pipelines. Its feasibility is verified through a field experiment. However, the novelty of the PPR inspection need more explanations. Why we need three channels? What are the benefits and how to verify them? Below are other detailed comments.

1.        The research objective of this study is not clearly demonstrated in the introduction section.

2.        The technical specifications are repeatedly introduced in the whole manuscript. Please make them concise.

3.        The full names of the acronyms should be given for the first time.

4.        Why we need the antenna of 1.4 GHz. What is its detection target? Why it is on the top?

5.        Can the robot measure GPR signal in the circumferential direction?

6.        The parameter settings during data acquisition should be detailed.

7.        The center frequency of antenna should be corrected to 1.4 GHz in Table 1 (Page. 7).and Line 211(Page 8).

8.        What is the size of the pipeline under test. More information about the field test should be given.

9.    Due to the humid environment inside the pipeline, the PPR may slip or rotate during the detection process. How can this issue be avoided?

10.    Please show the arrangement of ranging wheels onboard the designed PPR. The position of GPR antenna is measured by the odometer?

There are numerous typos and errors in the manuscript. A thorough proofreading by a native English speaker or ChatGPT is recommended.

Author Response

We feel great thanks for your professional review work on our article. As you are concerned, there are several problems that need to be addressed. According to your nice suggestions, we have made extensive corrections to our previous draft, the detailed responses to you are listed below.

Comment: In this manuscript, a three-channel pipe penetrating radar (PPR) is designed for inspection of the structural and functional conditions of underground drainage pipelines. Its feasibility is verified through a field experiment. However, the novelty of the PPR inspection need more explanations. Why we need three channels? What are the benefits and how to verify them?

Response: The three-channel pipe penetrating radar (PPR) antenna format was chosen to detect defects inside and outside the pipeline from multiple angles and to enhance the comprehensiveness of the pipeline inspection. In comparison to current conventional Closed-Circuit Television (CCTV) pipeline inspection technology, our device has the advantage of being able to detect defects in the external space of the pipe wall, such as void type and soil anomalies, without interrupting pipeline operation. In comparison to single-channel pipeline inspection robots with ground penetrating radar (GPR) technology, the advantages of our design are the multi-angle detection range and the greater depth of detection. The above expression is in our revised manuscript.

Point 1: The research objective of this study is not clearly demonstrated in the introduction section.

Response 1: Thanks for your suggestion. We add in the introductory chapter a comparison of our device with existing CCTV technology and single-channel pipeline radar, making it clear that the research objectives of the research and development of three-channel pipeline radar are to be able to exhaustively detect internal and external defects in urban drainage pipes using non-destructive techniques, to detect corrosion in the pipes, to determine the remaining thickness of the pipes, cavities outside the walls and soil anomalies, and to provide a basis for the timing of rehabilitation work, thereby reducing the risk posed by pipeline leaks.

Point 1: The technical specifications are repeatedly introduced in the whole manuscript. Please make them concise.

Response 2: We sincerely thank you for careful reading. According to your suggestion, we present the technical specifications in a concise manner in a revised manuscript.

Point 3: The full names of the acronyms should be given for the first time.

Response 3: We feel sorry for our carelessness. In our revised manuscript, the full names of the acronyms were given for the first time. Thanks for your correction.

Point 4: Why we need the antenna of 1.4 GHz. What is its detection target? Why it is on the top?

Response 4: We found that the corrosion at the top of the pipe is the most serious according to the relevant research, based on the needs of the detection accuracy of the 1.4GHz antenna for the detection of corrosion at the top of the pipe to determine the thickness of the pipe wall. We have added this section to Page 3 of the revised manuscript.

Point 5: Can the robot measure GPR signal in the circumferential direction?

Response 5: Yes, we designed the robot with a set of McNamee wheels mounted on top as support wheels for the curved radar antenna, which allows the robot to spiral forward inside the pipe and thus measure the GPR signal in the circumferential direction. We have added this section to Page 8 of the revised manuscript.

Point 6: The parameter settings during data acquisition should be detailed.

Response 6: Thanks for your suggestion. We have added a description of the parameters to be set for the three-channel pipeline radar data acquisition in the application section. The 1.4GHz radar antenna is used to detect corrosion depth and wall thickness at the top of the pipeline with a sampling time window of 10ns. 750MHz radar antenna is used to detect voids on the outside of the pipeline with a sampling time window of 30ns. The instrument is triggered by a high-precision rangefinder wheel in the line trolley with a trace interval of 0.5 mm. We have added this section to Page 10 of the revised manuscript.

Point 7: The center frequency of antenna should be corrected to 1.4 GHz in Table 1(Page.7).and Line 211(Page 8).

Response 7: We feel sorry again for our carelessness. In our revised manuscript, the mistake is revised. Thanks for your correction.

Point 8: What is the size of the pipeline under test. More information about the field test should be given.

Response 8: The pipes we tested in the field were concrete drainage pipes with a burial depth of 3m and a diameter of 400-600 mm. We have added this section to Page 10 of the revised manuscript.

Point 9: Due to the humid environment inside the pipeline, the PPR may slip or rotate during the detection process. How can this issue be avoided?

Response 9: We designed the PPR with a pair of guide wheels mounted on the top forward direction. When the PPR rotates, the inertial navigation sensors inside the PPR acquire the angle of rotation and the master control chip controls the reverse rotation of the guide wheels to make the PPR correct the deflection. We have added this section to the implementation of the device in the revised manuscript (Line 337~341, Page 8) and have labelled this guide wheel structure in Figure 5 (Page 7). The control algorithm for correcting its own rotation will be improved in the next step of development to improve or avoid this problem.

Point 10: Please show the arrangement of ranging wheels onboard the designed PPR. The position of GPR antenna is measured by the odometer?

Response 10: As the robot travels through the tube, the distance measuring wheel and encoder fixed to the cable are rotated by the transmission cable to obtain mileage information. The ranging wheel is shows in Figure 8 (Page 10).

We tried our best to improve the manuscript and made some changes marked up using the “Track Changes” function in revised manuscript. We appreciate for your warm work earnestly, and hope the correction will meet with approval. Once again, thank you very much for your comments and suggestions.

Reviewer 2 Report

The paper describes the design and development of a three-channel drainage pipeline ground penetrating radar (GPR) inspection device to improve the detection accuracy and efficiency of drainage pipeline problems. The device was tested on the actual sewage pipeline in Beijing, China, resulting in the discovery of 87 defects. The proposed device is working effectively. However, authors provide no comparison, and does not discuss the alternative devices. It is very difficult to discover the importance of the research and significance of content. I have written several drawbacks, which in my opinion must be improved:

Title. In my opinion title of the article should be slightly modified. There is curious wording used by the authors, such as: Design and Implementation of […] Radar Detection Device. Detection system and Radar in a very broad sense has similar tasks (not regarding technologies behind). Therefore, I suggest updating the title of the paper to avoid tautology.

Abstract. Is well written and is easy to read. Main contributions are declared by the authors.

Introduction. In line 43. Authors are declaring problems of existing methods. I believe several references and short discussion about disadvantages and advantages of currently employed systems (or systems which authors mention as having limitations) would be interesting for the reader. For example in Line 56 authors give several statements about typical detection equipment. Introduction chapter should briefly discuss such systems, and their main parameters.

In the introduction chapter it is not fully identified from where introduced problems originate. Thus, it is not clear what authors are trying to solve.

I am not sure what is "pipeline disease", or “detection of drainage pipeline disease engineering”. Is the term used in similar research?

Design. In Figure 3 direction of arrows, connecting components are unclear.

Application. Results of found defects are provided, however there are no references regarding the single-channel approaches, which authors claim to improve over. It is not possible to reach the conclusion, that proposed device is advantageous over the alternatives. Authors reach the conclusions, that accuracy and efficiency was improved. How was the efficiency and accuracy measured and how it was compared?

Additionally, authors should discuss the designed device with existing devices, showing the advantages and disadvantages of each.

Conclusions.

As I mentioned above, it is difficult to validate conclusion 1. Since authors does not properly show, that similar three-channel radar does not exist, or the drawbacks of single-channel radars.

Conclusion no. 2 is technical characteristics of the designed prototype. Authors should briefly discuss the advantages or challenges of the proposed design.

Conclusion no. 3 is statement of the achieved results, and it is not a conclusion. Authors should draw conclusion from the obtained results.

Small comments.

GPR should be defined in text (even if it is defined in abstract)

Not clear sentence meaning in line 40 “Robotic for water main in pipe inspection, in-pipe robot localization”

Missing references 17-18.

There are some spacing issues (i.e. Caption of chapter 2; Line 34). Text should be fully reviewed for similar errors. 

Quality of English is good. There are few comments, which I stated above. Biggest issue is about using the term "pipeline disease".

Author Response

We feel great thanks for your professional review work on our article. As you are concerned, there are several problems that need to be addressed. According to your nice suggestions, we have made extensive corrections to our previous draft, the detailed responses to you are listed below.

Comment: The paper describes the design and development of a three-channel drainage pipeline ground penetrating radar (GPR) inspection device to improve the detection accuracy and efficiency of drainage pipeline problems. The device was tested on the actual sewage pipeline in Beijing, China, resulting in the discovery of 87 defects. The proposed device is working effectively. However, authors provide no comparison, and does not discuss the alternative devices. It is very difficult to discover the importance of the research and significance of content.

Response: The three-channel drainage pipeline ground penetrating radar (GPR) inspection device to detect defects inside and outside the pipeline from multiple angles and to enhance the comprehensiveness of the pipeline inspection. In comparison to current conventional Closed-Circuit Television (CCTV) pipeline inspection technology, our device has the advantage of being able to detect defects in the external space of the pipe wall, such as void type and soil anomalies, without interrupting pipeline operation. In comparison to single-channel pipeline inspection robots with ground penetrating radar (GPR) technology, the advantages of our design are the multi-angle detection range and the greater depth of detection. The above expression is in our revised manuscript.

Point 1: Title. In my opinion title of the article should be slightly modified. There is curious wording used by the authors, such as: Design and Implementation of […] Radar Detection Device. Detection system and Radar in a very broad sense has similar tasks (not regarding technologies behind). Therefore, I suggest updating the title of the paper to avoid tautology.

Response 1: Thanks for your suggestion. We have amended the title to “Design and Implementation of Three-channel Drainage Pipe-line Ground Penetrating Radar Device” to avoid tautology.

Point 2: Abstract. Is well written and is easy to read. Main contributions are declared by the authors.

Response 2: Thank you for your confirmation. We will improve the problems in other chapters in the revised manuscript.

Point 3: Introduction. In line 43. Authors are declaring problems of existing methods. I believe several references and short discussion about disadvantages and advantages of currently employed systems (or systems which authors mention as having limitations) would be interesting for the reader. For example, in Line 56 authors give several statements about typical detection equipment. Introduction chapter should briefly discuss such systems, and their main parameters.

Response 3: Thanks to your suggestion, we have added to the introduction section of the revised manuscript, a discussion of existing drain inspection techniques such as CCTV and GPR techniques. We compare the key parameters between this system and the other GPR systems (ACPS) in Table 2 (Page 9).

Point 4: In the introduction chapter it is not fully identified from where introduced problems originate. Thus, it is not clear what authors are trying to solve.

Response 4: Thanks to your suggestions, we have clearly attempted to solve the problem that CCTV technology can only detect corrosion, cracks and filling conditions on the surface of the pipe, but not external defects in drainage pipes, in the introduction section of the revised manuscript. There is also the issue of low depth and coverage of pipeline inspection equipment with single-channel ground penetrating radar technology.

Point 5: I am not sure what is “pipeline disease", or “detection of drainage pipeline disease engineering”. Is the term used in similar research?

Response 5: We apologized for any confusion caused by our improper expressions. In our revised manuscript we have changed the term “pipe disease “, or “detection of drainage pipeline disease engineering” to “pipe defects” and” detection of drainage pipeline defect engineering”.

Point 6: Design. In Figure 3 direction of arrows, connecting components are unclear.

Response 6: Thanks to your corrections, we have redrawn Figure 3 to clearly arrows pointing and connecting components.

Point 7: Application. Results of found defects are provided, however there are no references regarding the single-channel approaches, which authors claim to improve over. It is not possible to reach the conclusion, that proposed device is advantageous over the alternatives. Authors reach the conclusions, that accuracy and efficiency was improved. How was the efficiency and accuracy measured and how it was compared?

Response 7: Thanks to your correction, we have added a comparison of the key parameters of this device and the single-channel pipeline radar device to the revised manuscript (Page 9) in order to draw conclusions for improvement.

Point 8: Additionally, authors should discuss the designed device with existing devices, showing the advantages and disadvantages of each.

Response 8: Thanks to your corrections, we have added a discussion of this device and existing devices in the introduction section (Page 2) of the revised manuscript, showing the advantages and disadvantages of each device.

Point 9: Conclusions. As I mentioned above, it is difficult to validate conclusion 1. Since authors does not properly show, that similar three-channel radar does not exist, or the drawbacks of single-channel radars.

Conclusion no.2 is technical characteristics of the designed prototype. Authors should briefly discuss the advantages or challenges of the proposed design.

Conclusion no.3 is statement of the achieved results, and it is not a conclusion. Authors should draw conclusion from the obtained results.

Response 9: Thanks to your suggestions, we have rewritten the conclusions section (Page 12) in the revised manuscript to highlight the advantages of the improvements to this device and the conclusions drawn from the results.

Small comments.

Point 11: GPR should be defined in text (even if it is defined in abstract)

Response 11: Thanks for your suggestions, we have explained the definition of GPR in the introductory section.

Point 12: Not clear sentence meaning in line 40“Robotic for water main in pipe inspection, in-pipe robot localization”

Response 12: We apologized for any confusion caused by our use of a sentence whose meaning was unclear. We have corrected this error in our revised manuscript.

Point 13: Missing references 17-18. There are some spacing issues (i.e., Caption of chapter 2; Line 34). Text should be fully reviewed for similar errors.

Response 13: We were really sorry for our careless mistakes; we have corrected the missing labelling of references and the formatting issues in the revised manuscript and double-checked to avoid similar errors.

Point 14: Quality of English is good. There are few comments, which I stated above. Biggest issue is about using the term “pipeline disease".

Response 14: We apologized once again, we have improved the use of terminology and polish our article.

We tried our best to improve the manuscript and made some changes marked up using the “Track Changes” function in revised manuscript. We appreciate for your warm work earnestly, and hope the correction will meet with approval. Once again, thank you very much for your comments and suggestions.

Round 2

Reviewer 2 Report

Authors addressed all of my issues. Additionally, I see more positive improvements on the manuscript. Significance of the work is now identifiable. Future directions is also included.

Overall, English language is improved. There is some formatting errors in spacing of numbers and units; also on spacing citation references. There should be a space.